# Using a health belief model to assess COVID-19 vaccine intention and hesitancy in Jakarta, Indonesia

Irma Hidayana[1,2], Sulfikar Amir[3]*, Dicky C. Pelupessy[4], Zahira Rahvenia[2]

**1** Department Asian Studies/Public Health, St. Lawrence University, Canton, New York, United States of America, **2** LaporCovid19.org- Bona Indah Plaza A2-B11, Lebak Bulus, Jakarta, Indonesia, **3** Sociology Programme, School of Social Sciences, Nanyang Technological University, Singapore, Singapore, **4** Faculty of Psychology Universitas Indonesia, Kampus UI Depok, Jawa Barat, Indonesia

* sulfikar@ntu.edu.sg

**Data Availability Statement:** The data underlying this article are available in https://doi.org/10.21979/N9/2PPR4G.

## Abstract

Since January 2021, Indonesia has administered a nationwide COVID-19 vaccination. This study examined vaccine intention and identified reasons for vaccine hesitancy in the capital city of Jakarta. This is a cross-sectional online survey using the Health Belief Model (HBM) to assess vaccine intent predictors and describe reasons for hesitancy among Jakarta residents. Among 11,611 respondents, 92.99% (10.797) would like to get vaccinated. This study indicated that all HBM constructs predict vaccine intention (P< 0.05). Those with a high score of perceived susceptibility to the COVID-19 vaccine were significantly predicted vaccine hesitancy (OR = 0.18, 95% CI: 0.16–0.21). Perceived higher benefits of COVID-19 vaccine (OR = 2.91, 95% CI: 2.57–3.28), perceived severity of COVID-19 disease (OR: 1.41, 95% CI: 1.24–1.60), and perceived susceptibility of the current pandemic (OR = 1.21, 95% CI: 1.06–1.38) were significantly predicted vaccination intend. Needle fears, halal concerns, vaccine side effects, and the perception that vaccines could not protect against COVID-19 disease emerged as reasons why a small portion of the respondents (n = 814, 7.23%) are hesitant to get vaccinated. This study demonstrated a high COVID-19 vaccine intention and highlighted the reasons for vaccine refusal, including needle fears, susceptibility to vaccine efficacy, halal issues, and concern about vaccine side effects. The current findings on COVID-19 vaccination show that the government and policymakers should take all necessary steps to remove vaccine hesitancy by increasing awareness of vaccine efficacy and benefit interventions.

## Introduction

As of May 2021, Indonesia was one of the countries with the highest number of novel Coronavirus Diseases 2019 (COVID-19) cases with the lowest testing rate in Southeast Asia [1]. In December 2020, Indonesia received the first three million doses of the Sinovac-Coronavac vaccine [2]. Two days after the Indonesian authority of food and drug administration issued emergency use of Sinovac, the COVID-19 vaccination program was administered across the country with agglomerated urban areas as the main focus of vaccine distribution [3].

**Funding:** SA received funding from Singapore Ministry of Education Academic Research Fund Tier-1 Grant #RG53/19 to partially support this work. The funder did not play any role in preparation and data collection and analysis of the manuscript.

**Competing interests:** The authors have declared that no competing interests exist.

By the end of May 2021, Indonesia has reached 27 million shots of the COVID-19 vaccine. The Indonesian government has issued a regulation as a legal basis for vaccinating the Indonesian population prioritizing health care workers, older people, public servants, those with pre-existing medical conditions, and those who live in areas with high transmission of COVID-19 [4]. Despite the efforts of the Indonesian government to vaccinate as many people as possible, vaccine hesitancy has existed among the population [5]. At about the same time, 15 different vaccines were granted emergency use authorization by the World Health Organization (WHO) [6]. However, the China Sinovac-Coronavac was not yet approved for emergency use by WHO [7]. Many of those approved for use have high efficacy [6]. In different demographics, vaccine efficacy is an essential driver of vaccine uptake [8, 9].

As the capital city of Indonesia, with a population of around 10.56 million, Jakarta has been the epicenter of COVID-19 in Indonesia since the beginning of March 2020 [10]. On 30 March 2021, the capital city recorded 381,090 cases with a total number of deaths of 6,327 [11]. In addition, only 1,178,243 persons (39.3%) received the first dose out of 3,000,689 targets [11].

In the Southeast Asia region and worldwide, studies have been conducted to examine the intention of a vaccine against COVID-19. A previous study showed that COVID-19 vaccine uptake in Indonesia was influenced by the effectiveness of the vaccines [8]. Further, COVID-19 vaccine acceptance rates in ten lower-middle income countries in Asia, Africa, and South America were higher where vaccine safety and efficacy were high [12]. Similarly, vaccine hesitancy rates were low in Singapore [13] and New Zealand [14]. This paper assessed vaccine intention among the residents eligible to receive the shot and identified the specific reasons drawing hesitant attitudes towards COVID-19 vaccination during the first phase of the COVID-19 vaccination program in Jakarta.

## Health belief model as a theoretical framework

To pursue our goal, we applied the Health Belief Model (HBM) as the core framework in our study. As one of the most widely applied theories in health behaviors [15], the HBM consists of six domains that predict health behavior: perceived susceptibility, perceived severity, perceived benefits, perceived barriers, cues to action, and self-efficacy. As we studied one simple behavior, we excluded self-efficacy in this paper.

Perceived susceptibility refers to a belief about the possibility of getting a condition. This study addressed individuals' beliefs about getting impacted by two conditions: the COVID-19 pandemics and the vaccine. Within this construct, we studied individuals' perception of vaccine side effects, whether or not the vaccine could protect against infection, and halal concerns about the vaccine that may hinder individuals from getting vaccination against coronavirus infection. Perceived severity refers to feelings about the seriousness of having the COVID-19 disease. In a broader sense, we included severity related to social and financial consequences such as reduced income, loss of jobs, restricted family and social interactions. Moreover, as information and access to vaccination centers were found to be obstacles for some individuals [16], the construct of perceived barriers in this study is focused on technical aspects that individuals may have to access the vaccine. Perceived benefits refer to protection provided by COVID-19 vaccines. Finally, cues to action refers to a strategy or information source that promotes the adoption of a behavior [17].

## Methods

### Study participants and survey design

This cross-sectional study was performed from 30 April to 15 May 2021. Quota sampling was used to analyze data collected from the proportion of gender represented across the five

districts (West Jakarta, South Jakarta, East Jakarta, North Jakarta, and Central Jakarta) in the capital city, Jakarta. Data were collected through a web-based anonymous survey using a Qualtrics-based online questionnaire. The Jakarta Administration Bureau facilitated the distribution of questionnaires to the Jakarta population through JAKI, an application for administrative information for Jakarta residents. Inclusion criteria were that the respondents were Jakarta residents who were more than 18 years of age and with internet access, while those who work in Jakarta but live on the outskirts of the city were excluded from the study. The questionnaire was pilot tested and validated by local experts prior to the administration of the survey.

### Instruments

A 45-item structured questionnaire was developed to assess the study objectives. The survey consisted of questions that assessed demographic background (8 questions), health status and COVID-19 experience (3 questions), and HBM constructs (28 questions). A 5-point Likert scale (1 = strongly disagree to 5 = strongly agree) was used for the HBM portion of the questionnaire groups. Eight demographic variables were collected: gender, age, occupancy and whether the respondent works in the health area, their role in the local community, estimated monthly income, education level completed, and religious belief. Three questions assessed the comorbidities of the respondents and whether respondents and their families have existed or been diagnosed with COVID-19 (Yes/No). The survey was anonymous and contained no identifiable respondent information.

### Ethics statement

This study was approved by the Faculty of Psychology, Universitas Indonesia Research Ethics Committee in April 2021. Approval code: 039/FPsi.Komite Etik/PDP.04.00/2021/. The survey was conducted online. Informed consent was obtained before the respondent began participating in the study. Informed consent was documented on a digital platform. This study did not include minors.

### Statistical analysis

Descriptive statistics (mean, standard deviation, frequency) were obtained for all variables. The HBM-based statements were grouped according to their constructs (perceived susceptibility to the COVID-19 pandemic and the vaccine, perceived severity of COVID-19 disease, perceived barriers to vaccination, and perceived specific vaccine benefits, and cue to action). Cronbach's alpha was calculated for the constructs; see supplemental materials for the detailed values. Spearman's rho and Pearson Chi-Square correlation were used to assess the correlation between HBM construct and (1) demographic variables; (2) health status and COVID-19 experience variables. A logistic regression model was applied to examine HBM factors that significantly predicted COVID-19 vaccine intent and refusal. Additional regression test was done to study if COVID-19 health experience variables significantly predicted vaccination intention. All statistical analyses were performed using the IBM SPSS 26 software. A P-value of less than 0.05 (95 percent of confidence interval) was considered statistically significant in this study.

## Results

### Demographics characteristics

A total of 11,611 participants completed the survey. The study received proportional gender-based responses from all five districts within Jakarta province. As shown in Table 1,

**Table 1. Demographic characteristics of study participants.**

| Variable | Category (N = 11611) | n | (%) |
|---|---|---|---|
| Sex | Male | 5844 | 50.33 |
| | Female | 5767 | 49.67 |
| Age | 18–20 years old | 170 | 1.46 |
| | >20–30 years old | 1174 | 10.11 |
| | >30–40 years old | 2327 | 20.04 |
| | >40–50 years old | 3288 | 28.32 |
| | >50–60 years old | 2347 | 20.21 |
| | >60 years old | 2305 | 19.85 |
| Health-related jobs | Yes | 1378 | 11.87 |
| | No | 10233 | 88.13 |
| Occupation | Student | 197 | 1.70 |
| | Housewife | 4017 | 34.60 |
| | Educational Staff | 275 | 2.37 |
| | Doctor/midwife/nurse/other health workers | 99 | 0.85 |
| | Day laborer (on-line driver, street trader, etc) | 1136 | 9.78 |
| | Military/Police | 59 | 0.51 |
| | Business owner | 671 | 5.78 |
| | State worker | 252 | 2.17 |
| | Private worker | 2134 | 18.38 |
| | NGO worker | 44 | 0.38 |
| | Artist | 39 | 0.34 |
| | Unemployed | 1256 | 10.82 |
| | Other | 1432 | 12.33 |
| Role in local community | Youth leader | 1411 | 12.15 |
| | Woman leader | 1690 | 14.56 |
| | Religious leader | 328 | 2.82 |
| | Senior citizen | 3083 | 26.55 |
| | N/A | 5099 | 43.92 |
| Monthly income (Indonesian Rupiah) | < Rp. 2.500.000 | 6233 | 53.68 |
| | IDR 2.500.001- IDR 5.000.000 | 3697 | 31.84 |
| | IDR 5.000.001- IDR7.500.000 | 694 | 5.98 |
| | IDR 7.500.001-IDR 10.000.000 | 324 | 2.79 |
| | IDR 10.000.001-IDR 12.500.000 | 139 | 1.20 |
| | IDR. 12.500.001-IDR 15.000.000 | 116 | 1.00 |
| | > IDR 15.000.000 | 408 | 3.51 |
| Religion | Buddha | 214 | 1.84 |
| | Hindu | 45 | 0.39 |
| | Islam | 10168 | 87.57 |
| | Catholic | 468 | 4.03 |
| | Christian | 583 | 5.02 |
| | Indigenous Beliefs | 14 | 0.12 |
| | Other | 19 | 0.16 |
| | Do not answer | 100 | 0.86 |
| Education | Not finished elementary/Middle/High school | 2704 | 23.29 |
| | High School | 6119 | 52.70 |
| | Diploma/College/Post Graduate | 2788 | 24.01 |

(*Continued*)

**Table 1.** (Continued)

| Variable | Category (N = 11611) | n | (%) |
|---|---|---|---|
| Are currently being or had previously diagnosed with COVID-19 | Yes | 612 | 5.27 |
| | No | 10999 | 94.73 |
| Have family members who are currently being or had previously diagnosed with COVID-19 | Yes | 834 | 7.182 |
| | No | 10777 | 92.82 |
| Comorbidities | No | 8221 | 70.80 |
| | **Yes** | 3390 | 29.20 |
| | Cardiovascular Disease (CVD) | 270 | 2.33 |
| | Asthma | 290 | 2.50 |
| | Kidney Disease | 40 | 0.34 |
| | Diabetes Mellitus | 532 | 4.58 |
| | Hypertension | 1355 | 11.67 |
| | Autoimun | 31 | 0.27 |
| | Other | 473 | 4.07 |
| | Do not know | 833 | 7.17 |

approximately an equal number of females (49.67%) and males (50.33%) respondents, who were majority had a high school degree as their highest education level (52.70%, n = 6,119), were among those aged 40–50 old (28.32%), and 53.68% earned less than IDR 2.5 million (equal to USD 169) each month.

More than half of the respondents (62,45%) received their first dose of the COVID-19 vaccine. Only a small portion (29.2%) reported having chronic diseases. The majority of respondents (94.73%) and their families (52.79%) were not being and had not previously been diagnosed with COVID-19.

The survey revealed that only a small portion of the respondents was unwilling to get vaccinated (n = 814, 7.01%) and identified five factors describing such hesitancy. Almost two percent (1.73%) or 201 respondents showed a strong agreement of being afraid of needle injection, 2.5% (n = 290) strongly agreed that the available COVID-19 vaccine is not halal, 3.49% (n = 405) strongly agreed that the available vaccine does not provide protection from COVID-19 infection, and 3.62% (n = 420) were concerned about the vaccine side effects. In addition, 279 respondents (2.4%) expressed their concern that they were not included in the targeted vaccination population. See **Fig 1**.

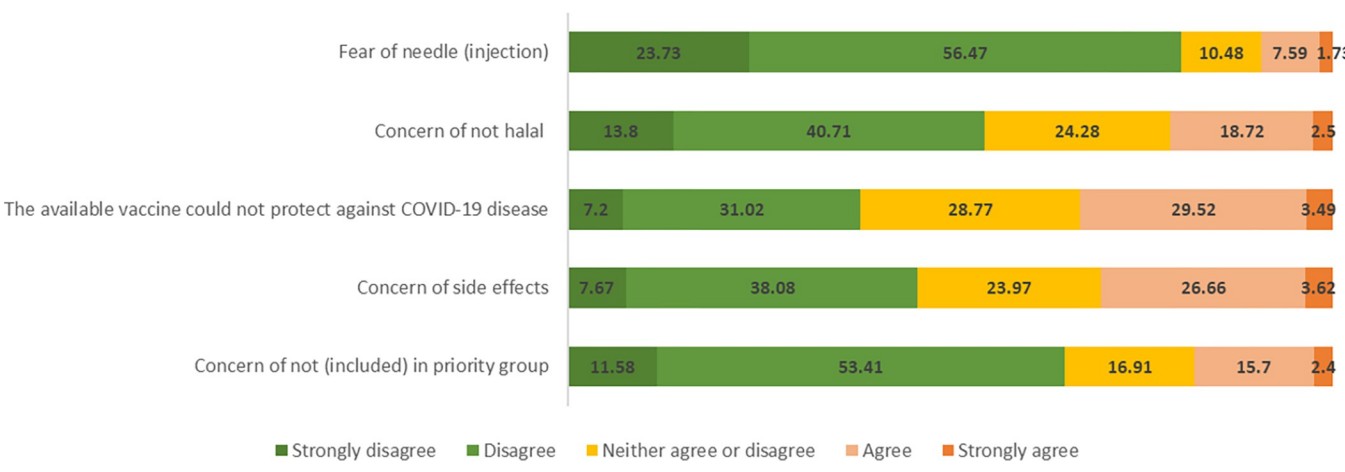

**Fig 1. Reasons for vaccine hesitancy.**

## Health beliefs and vaccine intention

Correlation coefficient analyses were used to examine the relationship between demographic variables and the HBM constructs and COVID-19 experience variables (Table 2). All

**Table 2. Correlation between the extended demography variables and the HBM constructs.**

| | Demographic variables | | Perceived susceptibility of COVID-19 pandemics | Perceived susceptibility of COVID19 vaccine | Perceived severity of COVID-19 disease | Perceived barriers to COVID-19 vaccine | Perceived benefits of COVID-19 Vaccine |
|---|---|---|---|---|---|---|---|
| Spearman's rho | Age | Correlation Coefficient | 0.022* | -0.133* | 0.004 | -0.039* | 0.046* |
| | | Sig. (2-tailed) | 0.016 | 0.000 | 0.638 | 0.000 | 0.000 |
| | Monthly income (Indonesian Rupiah) | Correlation Coefficient | 0.028* | -0.120* | 0.045* | -0.137* | 0.052* |
| | | Sig. (2-tailed) | 0.002 | 0.000 | 0.000 | 0.000 | 0.000 |
| | Highest Education level | Correlation Coefficient | 0.053* | -0.102* | 0.094* | -0.205* | 0.03 |
| | | Sig. (2-tailed) | 0.000 | 0.000 | 0.000 | 0.000 | 0.007 |
| Pearson Chi-Square | Sex | Contingency Coefficient | 0.068* | 0.106* | 0.084* | 0.047* | 0.057* |
| | | Sig. (2-tailed) | 0.000 | 0.000 | 0.000 | 0.000 | 0.000 |
| | Health-related jobs | Contingency Coefficient | 0.069* | 0.044* | 0.070* | 0.107* | 0.05* |
| | | Sig. (2-tailed) | 0.000 | 0.000 | 0.000 | 0.000 | 0.000 |
| | Occupation | Contingency Coefficient | 0.096* | 0.146* | 0.114* | 0.139* | 0.101* |
| | | Sig. (2-tailed) | 0.000 | 0.000 | 0.000 | 0.000 | 0.000 |
| | Role in Local community | Contingency Coefficient | 0.067* | 0.118* | 0.085* | 0.075* | 0.104* |
| | | Sig. (2-tailed) | 0.000 | 0.000 | 0.000 | 0.000 | 0.000 |
| | Religion | Contingency Coefficient | 0.125* | 0.173* | 0.119* | 0.107* | 0.103* |
| | | Sig. (2-tailed) | 0.000 | 0.000 | 0.000 | 0.000 | 0.000 |
| Pearson Chi-Square | Are currently being or had previously diagnosed with COVID-19 | Contingency Coefficient | 0.033* | 0.021 | 0.017 | 0.120* | 0.012 |
| | | Sig. (2-tailed) | 0.012 | 0.253 | 0.519 | 0.025 | 0.806 |
| | Family members are currently being or had previously diagnosed with COVID-19 | Contingency Coefficient | 0.022 | 0.021 | 0.042* | 0.022 | 0.021 |
| | | Sig. (2-tailed) | 0.214 | 0.297 | 0.000 | 0.250 | 0.288 |
| | Comorbidities | Contingency Coefficient | 0.051* | 0.163* | 0.048* | 0.023 | 0.06* |
| | | Sig. (2-tailed) | 0.000 | 0.024 | 0.000 | 0.186 | 0.000 |
| | COVID-19 Vaccination status | Contingency Coefficient | 0.037* | 0.272* | 0.073* | 0.128* | 0.139* |
| | | Sig. (2-tailed) | 0.003 | 0.000 | 0.000 | 0.000 | 0.000 |
| | Family consent to get vaccinated | Contingency Coefficient | 0.065* | 0.364* | 0.075* | 0.130* | 0.282* |
| | | Sig. (2-tailed) | 0.000 | 0.000 | 0.000 | 0.000 | 0.000 |
| | Vaccination willingness | Contingency Coefficient | 0.060* | 0.374* | 0.073* | 0.118* | 0.282* |
| | | Sig. (2-tailed) | 0.000 | 0.000 | 0.000 | 0.000 | 0.000 |

*. Correlation is significant at the 0.05 level (2-tailed).

**Table 3. Regression analysis—predictors of the HBM construct and COVID-19 vaccine intention.**

| Explanatory variable | β | Standard Error | P-Value | Wald Test | Odds Ratio OR | 95% CI of OR | |
|---|---|---|---|---|---|---|---|
| | | | | | | Lower | Upper |
| Perceived susceptibility to COVID-19 pandemic | 0.2 | 0.07 | 0.004 | 9.31 | 1.21 | 1.06 | 1.38 |
| Perceived susceptibility to COVID-19 vaccine | -1.71 | 0.07 | 0 | 686.5 | 0.18 | 0.16 | 0.21 |
| Perceived severity of the COVID-19 disease | 0.34 | 0.06 | 0 | 30.46 | 1.41 | 1.24 | 1.6 |
| Perceived barriers to vaccination | -0.15 | 0.06 | 0.006 | 8.78 | 0.85 | 0.77 | 0.96 |
| Perceived benefits of COVID-19 vaccine | 1.07 | 0.06 | 0 | 289.49 | 2.91 | 2.57 | 3.28 |
| Constant | 2.47 | 0.23 | 0 | 121.57 | 13 | | |
| **Test** | **Categories** | $\chi^2$ | **DoF** | **P-Value** | | | |
| Overall Model Evaluation | Likelihood Ratio Test | 1686.46 | 5 | 0 | | | |
| Goodness of Fit Test | Hosmer and Lemeshow Test | 8.48 | 8 | 0.38 | | | |

demographic variables except age were significantly correlated with the respondents' perceived susceptibility to the COVID-19 pandemic and the vaccine, perceived severity of the COVID-19 disease, perceived barriers to vaccination, and perceived specific vaccine benefits (P< 0.05). Age was not associated with the respondents' perceived severity of the COVID-19 disease. **Table 2** details the relationship between whether respondents and their families were being or had previously been diagnosed with COVID-19, respondents' comorbidities, and the HBM construct. In addition, more than half of the respondents had received vaccination during the survey (n = 7,251, 62.45%). Table 2 details the relationship between whether respondents and their families were being or had previously been diagnosed with COVID-19, respondents' comorbidities, and the HBM construct.

This study indicated that all HBM construct predicts vaccine intention (P < 0.05) as described in **Table 3**. Those with a high score of perceived susceptibility or concern to COVID-19 vaccine (OR = 0.18, 95% CI: 0.16–0.21, P < 0.05) and perceived technical barriers (OR = 0.85, 95% CI: 0.77–0.96, P< 0.05) were less likely to get vaccinated than those with less scores. Perceived higher benefits of COVID-19 vaccine (OR = 2.91, 95% CI: 2.57–3.28, P < 0.05), perceived severity of the COVID-19 disease (OR: 1.41, 95% CI: 1.24–1.60, P < 0.05), and perceived susceptibility of the current pandemic (OR = 1.21, 95% CI: 1.06–1.38, P < 0.05) were significantly more likely to intend to get vaccinated. Two variables in respondents and family health conditions are found to be determinant factors for vaccine intent and hesitance as described in **Table 4**. Respondents with family members who were being or had previously been diagnosed with COVID-19 were one and a half times more likely to get vaccination than those who were not being or previously had not been diagnosed with COVID-19 (OR = 1.50, 95% CI: 1.09–2.05, P < 0.05). In addition, those who have comorbidities were less

**Table 4. Regression analysis—predictors of extended demography variables and COVID-19 vaccine intentions.**

| Explanatory variable | β | Standard Error | P-Value | Wald Test | Odds Ratio OR | 95% CI of OR | |
|---|---|---|---|---|---|---|---|
| | | | | | | Lower | Upper |
| Family members who are currently or have been diagnosed with COVID-19 (Yes) | 0.40 | 0.16 | *0.01* | 6.21 | 1.50 | 1.09 | 2.05 |
| Underlying medical condition (comorbidities) (Yes) | -0.680 | 0.07 | *0.00* | 85.75 | 1.98 | 1.72 | 2.29 |
| Constant | 2.12 | 0.06 | *0.00* | 1374.20 | 8.32 | | |
| **Test** | **Categories** | $\chi^2$ | **DoF** | **P-Value** | | | |
| Overall Model Evaluation | Likelihood Ratio Test | 87.691 | 2 | 0 | | | |
| Goodness of Fit Test | Hosmer and Lemeshow Test | 0.19 | 1 | 0.662 | | | |

likely to get vaccinated compared to those who have no comorbidities (OR = 0.50, 95% CI: 0.44–0.58, P < 0.05).

## Discussion

The present study examined vaccine intention and described reasons for vaccine hesitancy vis-a-vis vaccine acceptance among Jakarta residents. This was conducted during the first phase of the COVID-19 vaccination rollout in Jakarta in which COVID-19 cases and deaths were the highest in the nation. By 30 April 2021, at the start of the present study, Jakarta recorded 408,620 cases, of which 6,733 died from COVID-19 [10]. Hence, the local health authority delivered the vaccination program massively and rapidly.

At the time of the study, vaccination priority was given to health care workers, older citizens, and those who work in public service areas [4]. Jakarta has targeted 3,000,689 people to receive COVID-19 vaccination during the first phase of vaccine rollout. As of 30 April 2021, 1,906,096 or 63.5% of the target population have received the first dose, while 1,214,494 or 40.5% have completed the second dose [10]. While the sample in this study comprises more of the non-priority vaccine population, when we looked into the respondents' occupations, we found nearly 12% (n = 1,378) had healthcare-related jobs and about forty percent of the respondents (n = 4,652, 40.06%) were elderly. The findings are in-line with the vaccination statistic data, for which 62.45% or 7,251 respondents have received the first dose.

Consistent with the previous studies [8, 18–20], the respondents in our study demonstrated positive intention toward COVID-19 vaccination (n = 10,797, 92.77%). This substantial proportion of positive intention toward COVID-19 vaccination exceeds the finding that 96 countries achieved lower than the WHO target of 40% of vaccination coverage by the end of 2021 and that lower-middle-income countries could achieve between 28% to 80% vaccination coverage [21].

However, although only a small portion of the respondents (n = 814, 7.01%) was unwilling to uptake the COVID-19 vaccine, scrutinizing the reasons for vaccine hesitancy helps better understand the barriers and formulate recommendations, especially communication to address the obstacles. Addressing vaccination barriers in Jakarta, the COVID-19 epicenter of Indonesia, is critical to ensure most of its population is protected by vaccines. This present study excavated five reasons as such barriers to vaccine hesitancy. One of the barriers this study revealed is needle fears or being afraid of injection among 201 respondents (1.73%). This is not surprising because the previous study demonstrated that some Indonesian adults are afraid to inject a needle into the body [22]. In fact, fear of needle injection has been recognized in healthcare areas. Despite needle fears being common among children, a study in the USA estimated that 11.5 to 66 million U.S. adults might encounter this condition [23]. Consequently, this group often avoids seeking medical care which may lead to vaccination refusals [24].

Moreover, consistent with Baraniuk [25] and Singh and Upshur [26], 290 respondents (2.5%) believe that the available COVID-19 vaccine is not halal, which led them to refuse to get the vaccination. It should be noted that China Sinovax's Coronavax was the only vaccine available when the survey was conducted. As the most populous Muslim country, religious consideration, including a halal certification of a vaccine, is critical for vaccine acceptance [27]. The halal issue on vaccination has existed in the nation and is attributable to vaccine refusal [28, 29]. For example, a previous study has demonstrated a sharp decline in the measles and rubella vaccination when the population doubts whether the vaccine qualifies as halal [30]. Therefore, as a Muslim majority population, it is critical to issue halal certification as soon as the Emergency Use Authorization of a vaccine is announced to reduce refusal.

In addition to the halal issue, we found 405 or 3.49% of respondents perceived that the available vaccine could not protect against COVID-19 disease. This finding has been consistent with the most recent study that assesses vaccine acceptance. Harapan et al. [8] indicated that the likelihood of people to uptake vaccination is if the COVID-19 vaccine had 95% efficacy. Thus, this present study underlined the importance of higher effectiveness perceived as efficacy could impact vaccine uptake. Next is the concern about vaccine side effects (n = 420, 3.65%), which is consistent with findings from numerous studies [9, 19, 31, 32]. Wong et al. [32] indicated that fears about vaccine adverse effects are indicated as among the strongest barriers to vaccination, which was described by having heard of adverse effects after vaccination and having heard of death cases after vaccination. The last reason for vaccine hesitancy we found in this study is the concern of not being included in the vaccination program (n = 279, 2.4%). As described previously, the local authority only inoculated COVID-19 vaccination for the most vulnerable target population to protect against COVID-19 [4]. Thus, such concern is reasonable amidst limited available vaccine stock, and vaccination for a wider public is yet to be available.

Furthermore, although our findings suggest that people's beliefs or perceptions about the susceptibility and severity of current COVID-19 pandemics and vaccines, including perceived benefits and technical barriers to access vaccines, were determinants of vaccine intent or refusal, greater attention should be emphasized to the perceived vaccine susceptibility (B = -1.72, $P < 0.05$) and the benefit of the vaccine to protect against COVID-19 (B = 1.023, $P < 0.05$). In this study, these two variables significantly provided major contributions to predicting vaccine intention and refusal compared to the other HBM variables. Again, these findings highlighted the pivotal role of removing barriers to halal issues and fears of needle injection. Moreover, the effectiveness of vaccines was one of the essential drivers for vaccine uptake [8, 9]. Therefore, as this study suggested, ensuring to provide vaccines with a higher efficacy level is more likely to reduce vaccine hesitancy.

Lastly, this study indicated that self and family health conditions significantly predicted vaccine intention. Those who have comorbidities were less likely to get vaccinated compared to those who have no comorbidities (OR = 0.50, 95% CI: 0.44–0.58, $P < 0.05$). These findings are in line with studies conducted elsewhere: in Northern Italy [33] and Brazil [34]. The results of this study are consistent with others showing that COVID-19 vaccine hesitancy is more common among people with comorbidities.

## Conclusion

This study demonstrated a high COVID-19 vaccine intention (n = 10,797, 92.77%). Four major factors have been identified as predictors of such high uptake, i.e., perceived COVID-19 disease susceptibility (OR = 1.34, P = 0.00), the technical barrier to access vaccination (OR = 0.58, P = 0.00), family members who were currently being or previously had diagnosed with COVID-19 (OR = 1.42, P = 0.03), and self-comorbidities (OR = 1.89, P = 0.00). Additionally, this study underscored the importance of identifying reasons for vaccine refusal. Needle fears, susceptibility to vaccine efficacy, halal issues, concern about vaccine side effects and comorbidities, and not being included in the vaccination targeted group were indicated as barriers to vaccine uptake. Although only accounted for by a small number of respondents, it is plausible to address these specific barriers, given that Jakarta always had the highest COVID-19 cases and deaths. This study suggests that education on vaccine efficacy and benefit interventions, which encompasses removing vaccine hesitancy, is critically needed to promote vaccine uptake. Lastly, there is a need for further similar studies in the same population that might provide a comprehensive picture of vaccination intentions and barriers.

## Limitations

This study has two limitations. First, we used a simple stratification of the sample based on the sample's gender proportion. However, the quota sampling employed could lead to sampling bias because the sample has not been chosen using random selection. The generalizability of the survey results may be impacted by how we distributed the online questionnaire. Second, the Jakarta administration team helped us to disseminate the questionnaire using an application for Jakarta residents. As a result, it might not reach people with no internet and no smartphone access, thus affecting data representation.

## Author Contributions

**Conceptualization:** Irma Hidayana, Sulfikar Amir, Dicky C. Pelupessy.

**Data curation:** Irma Hidayana, Zahira Rahvenia.

**Funding acquisition:** Sulfikar Amir.

**Methodology:** Irma Hidayana, Dicky C. Pelupessy.

**Validation:** Zahira Rahvenia.

**Writing – original draft:** Irma Hidayana, Sulfikar Amir.

**Writing – review & editing:** Irma Hidayana, Sulfikar Amir, Dicky C. Pelupessy.

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
