## [Decision Letter · Decision Letter 0]

21 Jun 2022

PGPH-D-22-00144

Using a Health Belief Model to Assess COVID-19 Vaccine Intention and Hesitancy in Jakarta, Indonesia

Dear Dr. Amir,

Thank you for submitting your manuscript to PLOS Global Public Health. After careful consideration, we feel that it has merit but does not fully meet PLOS Global Public Health’s publication criteria as it currently stands. Therefore, we invite you to submit a revised version of the manuscript that addresses the points raised during the review process.

Please submit your revised manuscript by . If you will need more time than this to complete your revisions, please reply to this message or contact the journal office at globalpubhealth@plos.org. Please include the following items when submitting your revised manuscript:

We look forward to receiving your revised manuscript.

Kind regards,

Mahbub-Ul Alam, MPH

Academic Editor

Journal Requirements:

State the initials, alongside each funding source, of each author to receive each grant.

2. Please update your online Competing Interests statement. If you have no competing interests to declare, please state: “The authors have declared that no competing interests exist.”

3. Please provide separate figure files in .tif or .eps format and remove any figures embedded in your manuscript file. Please also ensure that all files are under our size limit of 10MB.

4. Please note that your Data Availability Statement is currently missing the direct link to access each database as we cannot access the provided link. If your manuscript is accepted for publication, you will be asked to provide these details on a very short timeline. We therefore suggest that you provide this information now, though we will not hold up the peer review process if you are unable.

Additional Editor Comments (if provided):

Reviewers' comments:

Reviewer's Responses to Questions

**Comments to the Author**

1. Does this manuscript meet PLOS Global Public Health’s publication criteria? Is the manuscript technically sound, and do the data support the conclusions? The manuscript must describe methodologically and ethically rigorous research with conclusions that are appropriately drawn based on the data presented.

Reviewer #1: Yes

Reviewer #2: Yes

2. Has the statistical analysis been performed appropriately and rigorously?

Reviewer #1: Yes

Reviewer #2: Yes

3. Have the authors made all data underlying the findings in their manuscript fully available (please refer to the Data Availability Statement at the start of the manuscript PDF file)?

Reviewer #1: Yes

Reviewer #2: No

4. Is the manuscript presented in an intelligible fashion and written in standard English?

Reviewer #1: Yes

Reviewer #2: Yes

5. Review Comments to the Author

Reviewer #1: Congratulations to the research team for their thoughtful manuscript. The statistical analysis has been accurately done and presented and the findings either provides new information or consistent with other findings.

My only specific comment is the need to standardize how numbers are captured in the manuscript. For instance: the use of commas or periods in numbers/figures wasn't consistent such as IDR 10.000.001-IDR 12.500.000 (use of period in table 1 for income) or in the discussion you notice a number captured as 1,906,096 (with commas) or 11611 in table 1 (without period or commas). Same for P-values: where decimal is used (P< .05) but in the tables you see P-values reflected as 0,006 (with commas). In the discussion you have for instance n= 4,652, 40.06% (decimals used) but in the tables you see percentages using commas (not decimals for consistency). The lack of consistency is my challenge.

Recommendation: In as much as how the numbers have been reflected by the authors is not wrong, there is need for standardization and uniformity in the use of periods and commas in the tables and in the main write up to make it easier for the reader to follow. e.g either use P<0.05 or P<0,05 for consistency or 1,100,000 or 1.100.000 for consistency and not a mixture of the two formats.

Reviewer #2: Thank you for inviting me to review " Using a Health Belief Model to Assess COVID-19 Vaccine Intention and Hesitancy in Jakarta, Indonesia"

Comments # 1: Abstract

Line number 43: Please add 0 (zero) before writing a decimal number and add a space between CI:2.57

After line 51, the authors should give a conclusion or solution-related line that beautifies the abstract part. The author can mention something like the following line. For example, the current findings on COVID-19 vaccination show that the government and policymakers should take all necessary steps to ensure the effectiveness of the vaccination program

Comments # 2: Introduction

Line number 87, 88: Correct the statement of only 1.178.243 persons (39.3%) received the first dose out of 3.000.689 targets. This indicates an inaccurate number, please correct the right number as 1,178,243 persons and 3,000,689 targets in place of 1.178.243 persons and 3.000.689 targets

The author should include some references that have summarized vaccine acceptance and vaccine hesitancy. The author also ought to include some studies that have been conducted in developing countries like Asia and developed countries to support this manuscript

Comments # 3: Methods

Study participants and survey design

Line number121: What is the name of the 5 districts? Please included

Line number 125: What are the exclusion criteria of this survey? What were the reason for only adding inclusion criteria?

Instruments:

Line number 130: What types of questionnaires were used in this study? Was it structured, or semi-structured or other types?

Statistical analysis

Why is not mentioned confidence interval (CI) in the statistical analysis section

Comments # 4: Results

Line number 168: use space between 2.5million

Table 1 Headings section: Please write n instead of count and % in lieu of Percentage (%)

Correct the proper percentage of table 1 in each percentage row. This column shows the inaccurate percentage and uses the full stop (.) symbol in place of a comma (,)

It is not essential to show the total number of respondents for each categorical row. It can be shown in the table 1 headline as

variable category Overall respondent N= 11611

n %

Sex Male 5844 50.33

Remove highlights in the rows of (Religion Do not answer 100)

Use single space in the row of (Have family members who are currently being or had previously diagnosed with COVID-19 No 10777 92,80). This row contains double space

Table 2

Similarly, it is not essential to show the total number of respondents for each categorical row. It can be written as like as table 1.

Why are not Spearman's rho and Pearson Chi-Square test analysis show different rows in a similar column? This analysis should be shown different rows in each column.

Line number 199: Write this sentence properly. Table 2 details the relationship between whether respondents and their families were being or had previously been diagnosed with COVID-19, respondents’ comorbidities, and the HBM construct

Table 3:

Include coefficient symbol instead of

Use full stop (.) in place of comma (,)

Check Table 3 last rows and correct this

Line number 236: Above line number 236, insert this table in proper format and use a proper Chi-Square symbol.

Table 4:

Include coefficient symbol in lieu of

Use full stop (.) instead of comma (,)

Check the whole Table 4. It seems to miss information in this table.

Line number 242: Below the line number 236, insert this table in proper format and use a proper Chi-Square symbol.

Comments # 5: Discussions

Line number 251: Insert this reference in similar format.

The author showed that 92.99% (10,797) would like to get vaccinated. But there is no sufficient explanation for it. The author also didn’t explain the overall world scenario of COVID-19 vaccine acceptance and hesitancy. So, the author should give some recent references to the related studies that clear the manuscript.

Line number 268: Address mathematical notation (%) in this line, we found nearly 12 percent

In lines 269, 270, and 271, this sentence doesn’t sound right and it should be written correctly.

Overall comments:

There are some grammatical errors as well as improving the English language. It is noticed that the structure of many sentences is misleading. Please revise the manuscript correctly, avoiding grammatical errors.

Please add 0 (zero) before writing a decimal number

Use full stop (.) in place of comma (,) for all table components were using a numerical number

6. PLOS authors have the option to publish the peer review history of their article (what does this mean?). If published, this will include your full peer review and any attached files.

**Do you want your identity to be public for this peer review?** For information about this choice, including consent withdrawal, please see our Privacy Policy.

Reviewer #1: No

Reviewer #2: **Yes: **Md. Zohurul Islam

---

## [Decision Letter · Decision Letter 1]

30 Sep 2022

Using a Health Belief Model to Assess COVID-19 Vaccine Intention and Hesitancy in Jakarta, Indonesia

PGPH-D-22-00144R1

Dear Dr Amir,

We are pleased to inform you that your manuscript 'Using a Health Belief Model to Assess COVID-19 Vaccine Intention and Hesitancy in Jakarta, Indonesia' has been provisionally accepted for publication in PLOS Global Public Health.

Best regards,

Ari Probandari, PhD

Academic Editor

Reviewer Comments (if any, and for reference):

Reviewer's Responses to Questions

**Comments to the Author**

1. If the authors have adequately addressed your comments raised in a previous round of review and you feel that this manuscript is now acceptable for publication, you may indicate that here to bypass the “Comments to the Author” section, enter your conflict of interest statement in the “Confidential to Editor” section, and submit your "Accept" recommendation.

Reviewer #1: All comments have been addressed

Reviewer #2: All comments have been addressed

2. Does this manuscript meet PLOS Global Public Health’s publication criteria? Is the manuscript technically sound, and do the data support the conclusions? The manuscript must describe methodologically and ethically rigorous research with conclusions that are appropriately drawn based on the data presented.

Reviewer #1: Yes

Reviewer #2: Yes

3. Has the statistical analysis been performed appropriately and rigorously?

Reviewer #1: Yes

Reviewer #2: Yes

4. Have the authors made all data underlying the findings in their manuscript fully available (please refer to the Data Availability Statement at the start of the manuscript PDF file)?

Reviewer #1: Yes

Reviewer #2: Yes

5. Is the manuscript presented in an intelligible fashion and written in standard English?

Reviewer #1: Yes

Reviewer #2: Yes

6. Review Comments to the Author

Reviewer #1: Congratulations for the efforts made in revising the manuscript based on the feedback.

Reviewer #2: (No Response)
